# Role of Ligand Distribution in the Cytoskeleton-Associated Endocytosis of Ellipsoidal Nanoparticles

**DOI:** 10.3390/membranes11120993

**Published:** 2021-12-19

**Authors:** Yudie Zhang, Long Li, Jizeng Wang

**Affiliations:** Key Laboratory of Mechanics on Disaster and Environment in Western China, Ministry of Education, College of Civil Engineering and Mechanics, Lanzhou University, Lanzhou 730000, China; zhangyd2012@lzu.edu.cn

**Keywords:** ligand distribution, cellular uptake, cytoskeleton, ellipsoidal nanoparticles

## Abstract

Nanoparticle (NP)–cell interaction mediated by receptor–ligand bonds is a crucial phenomenon in pathology, cellular immunity, and drug delivery systems, and relies strongly on the shape of NPs and the stiffness of the cell. Given this significance, a fundamental question is raised on how the ligand distribution may affect the membrane wrapping of non-spherical NPs under the influence of cytoskeleton deformation. To address this issue, in this work we use a coupled elasticity–diffusion model to systematically investigate the role of ligand distribution in the cytoskeleton-associated endocytosis of ellipsoidal NPs for different NP shapes, sizes, cytoskeleton stiffness, and the initial receptor densities. In this model, we have taken into account the effects of receptor diffusion, receptor–ligand binding, cytoskeleton and membrane deformations, and changes in the configuration entropy of receptors. By solving this model, we find that the uptake process can be significantly influenced by the ligand distribution. Additionally, there exists an optimal state of such a distribution, which corresponds to the fastest uptake efficiency and depends on the NP aspect ratio and cytoskeleton stiffness. We also find that the optimal distribution usually needs local ligand density to be sufficiently high at the large curvature region. Furthermore, the optimal state of NP entry into cells can tolerate slight changes to the corresponding optimal distribution of the ligands. The tolerance to such a change is enhanced as the average receptor density and NP size increase. These results may provide guidelines to control NP–cell interactions and improve the efficiency of target drug delivery systems.

## 1. Introduction

A thorough understanding of the interaction between nanoparticles (NPs) and cells is of significant interest. For instance, the effective design of NPs with penetration capability in target drug delivery systems is expected, and the safety and toxicity in extensive applications of industrial NPs [1,2,3] should also be of considerable concern. In the past decade, considerable effort has been focused to investigate the endocytosis of NPs depending on their size [4,5,6], shape [7,8,9,10,11,12], orientation [13,14,15], stiffness [16,17,18,19,20], cytoskeleton deformation [21,22,23,24], NP cooperativeness [25,26], membrane wrapping [27,28], cell matrix and substrate [29,30], and surface physicochemical properties [31,32]. Given the prevalent assumption that ligands are uniformly distributed, an open question about how ligand distribution may affect the internalization of NPs into cells is raised, despite the considerable progress mentioned above.

Recently, Schubertová et al. [33] focused on the specific case of receptors being overexpressed in cancer cells and demonstrated that spherical NPs with uniform ligand distribution yield the fastest wrapping speed by the cell membrane through performing a molecular dynamic simulation, in which the receptors can directly react with the ligands on the surface of NPs without diffusion. While the receptors have relatively low density, NP endocytosis is generally limited by receptor diffusion [4,8,34]. In the case of receptor-diffusion-mediated endocytosis of NPs, we have previously examined the effect of ligand distribution on wrapping cylindrical NPs by cell membrane and determined an almost uniform ligand distribution as the optimal distribution associated with the highest cellular uptake efficiency [35]. The above studies have focused on describing the effect of the ligand distribution on the internalizations of NPs with spherical and cylindrical shapes, both of which have surfaces of constant Gaussian curvature. In addition, cylindrical NPs have also been used to investigate how aspect ratios may affect the cell–NP interactions [7,36,37]. However, for NPs with complex shapes, it is still unclear how non-constant surface curvature may couple other factors like the distribution patterns of ligands and cytoskeleton remodeling to influence their cellular uptake.

Endocytosis includes a number of biophysical processes by which cells internalize NPs. It has been recognized that the cytoskeleton plays an essential role in several of these processes [38]. The molecular structure, on the basis of the association of the cytoskeleton to endocytosis, is made up of many endocytic proteins potentially linked to the actin cytoskeleton so that the NP entry via endocytosis essentially accompanies cytoskeleton remodeling [38,39,40,41,42,43,44]. Such a point of view has been confirmed through the observations of HIV viral particle entry into host cells, where a large number of protein–protein interactions are indicative of the links between the endocytic machinery and the actin cytoskeleton [45,46]. Most recently, while taking advantage of today’s nanotechnology, an experiment has been performed on measuring the dynamic process of internalization of a single human enterovirus 71 with size 26 nm via endocytosis by a force tracing technique on atomic force microscopy (AFM) [47]. It is found that if cytochalasin B, a cell-permeable mycotoxin that strongly inhibits network formation by actin filaments, is injected into the AFM liquid chamber during force tracing measurement, most of the force signal disappeared, which indicates that the wrapping of the NP cannot occur without the association of the actin filament network [47]. From a mechanics point of view, the involvement of cytoskeleton remodeling in the endocytic process embodies mechanical resistance through elastic deformation [21,22,23]. This deformation of the cytoskeleton during the NP–cell interaction, at the length scale of about ten nanometers, usually follows the linear theory of continuum mechanics, as confirmed by the experiments on nanoneedle (radius from 20 to 150 nm) indentation to living cells [48,49,50].

Although tremendous progress in understanding cell–NP interactions has been achieved, the role of ligand distribution in the cytoskeleton-associated endocytosis of non-spherical NPs remains unclear. In this work, we propose a coupled elasticity–diffusion statistical dynamic model to investigate the influence of ligand distribution on the cellular uptake of ellipsoidal NPs, where we assume that the uptake process is driven by the binding energy between diffusive receptors on the cell membrane and ligands on the NP surface to overcome resistances from membrane deformation, cytoskeleton deformation, and changes in the configuration entropy of receptors.

## 2. Materials and Methods

Figure 1 shows the schematic for the interaction between an ellipsoidal NP and a cell via cytoskeleton-associated endocytosis. The cell is modeled as an elastic half-space covered by a cell membrane embedded with diffusive mobile receptors. The ellipsoidal NP is considered as rigid and coated with immobile ligands. For simplicity, we consider the NP as an ellipsoid of revolution with equatorial and polar semi-axes, $R_{a}\;$and $R_{b}$, as shown in Figure 1a. We assume that the mobile receptors are distributed evenly with constant density $\xi_{0}$ prior to the engulfment. The ligand density $\xi_{L}\left(h\right)$, as a function of the central axis h, can be non-uniform. And the depth of the NP engulfment is denoted as $h_{d}$. As long as the receptors diffuse to binding sites and bind with the ligands on the NP surface, the receptor density within the contact area becomes identical to the ligand density. The distribution of the receptors on the membrane surface then become non-uniform, which is denoted as ξ(s,t) at time *t*. 

During the cytoskeleton-associated endocytosis of NPs, engulfment is driven by the formation of ligand-receptor complexes to overcome the resistance from the membrane and cytoskeleton deformations, and the changes in the configuration entropy of receptors. The energies that determine the internalization process include: (1) $F_{1}$, favorable binding energy of ligand–receptor; (2) $F_{2}$, unfavorable energy of membrane deformation; (3) $F_{3}$, unfavorable energy attributed to changes in the configuration entropy of receptors; and (4) $F_{4}$, unfavorable energy of cytoskeleton deformation. The total energy for cellular uptake of NPs can then be expressed by,
(1)$$F=F_{1}+F_{2}+F_{3}+F_{4}\;$$

The binding energy during the wrapping process can be obtained as,
(2)$$F_{1}=\int_{A_{w}}-e_{RL}\xi_{L}dA\;$$
where $A_{w}$ is the contact area between the NP and cell, and $e_{RL}$ is the chemical energy released by each receptor–ligand binding event. The axially symmetric system causes the infinitesimal area element to be expressed in the following form,
(3)dA=2πsds 

The fluid membrane theory of Canham–Helfrich [51] states that during NP–cell interaction, the energy of membrane deformation is,
(4)$$F_{2}=\int_{0}^{a\left(t\right)}2\pi s\left[2\kappa H^{2}\left(h\right)+\gamma \right]ds\;$$
where κ and γ are the bending modulus and surface tension of the membrane, respectively. Once the membrane is in close contact with the NP, its mean curvature, H, becomes equal to that of the corresponding position of the NP, and can be expressed as (see Appendix A for details),
(5)$$H\left(h\right)=\frac{\lambda }{2R_{b}}{\left(\left(2h/R_{b}-h^{2}/R_{b}^{2}\right)+\lambda^{2}{\left(1-h/R_{b}\right)}^{2}\right)}^{-\frac{3}{2}}+\frac{1}{2\lambda }{\left(2R_{b}h-h^{2}+\lambda^{2}{\left(h-R_{b}\right)}^{2}\right)}^{-1/2}$$
where $\lambda =R_{a}/R_{b}$.

To estimate the entropic contribution to the energy associated with the receptor distribution, we adopted the model based on the theory of ideal gas [4,52]. The unfavorable energy of changes in the configuration entropy of the bonds and free receptors is given as,
(6)$$F_{3}=\int_{0}^{a\left(t\right)}k_{B}T\left\{\xi_{L}\left(h\right)\ln\frac{\xi_{L}\left(h\right)}{\xi_{0}}\right\}2\pi sds+\int_{a\left(t\right)}^{\infty }k_{B}T\left\{\xi \ln\frac{\xi }{\xi_{0}}\right\}2\pi sds\;$$
where *T* is the absolute temperature in the order of 300 K.

In the cytoskeleton-associated endocytosis, the entry of NPs into the cells inevitably deforms the cytoskeleton. Obataya et al. [48] used an atomic force microscope with a long nanoneedle (radius = 100 nm to 150 nm) to perform indentations on living cells. They demonstrated that the loading force curve for an indentation depth of up to 2 µm remains consistent with the classic Hertz model in contact mechanics. Similarly, Beard et al. [49] used a nanoneedle probe with a radius of only 20 nm and recorded the loading force curve during the indentation of a corneocyte. The corneocyte is usually a target of numerous viruses, including the Herpes simplex virus Type 1 [53]. This study [49] confirmed that the Hertz model can well fit the loading force curve. As the Hertz model is derived based on the contact problem of two elastic bodies, these results therefore clearly indicate that living cells during indentation behave as a deformable elastic solid, implying that cytoskeleton deformation plays an important role in resisting NP intrusion. Based on the Hertz contact theory for rotating solids [54], the deformation energy of cytoskeleton can be derived as,
(7)$$F_{4}=8E^{*}\sqrt{R_{a}^{2}/R_{b}}h_{d}^{5/2}/15\;$$
where $R_{a}^{2}/R_{b}$ is the curvature radius at the initial contact point, and the combined elastic modulus is defined as $1/E^{*}=\left(1-\mu_{c}^{2}\right)/E_{c}+\left(1-\mu_{n}^{2}\right)/E_{n}$, in which $\mu_{c}$ and $E_{c}$ represent the Poisson ratio and Young modulus of the cell, and the corresponding values for NPs are $\mu_{n}$ and $E_{n}$. In the limit wherein NPs are stiffer than cells, the combined elastic modulus can be simplified to $E^{*}=E_{c}/\left(1-\mu_{c}^{2}\right)$.

The differentiation of free energy in Equation (1) with respect to time yields,
(8)$$\frac{dF}{dt}=-2\pi a\frac{da}{dt}\left\{e_{RL}\xi_{L}^{+}-\left(2\kappa H^{2}\left(h_{d}\right)+\gamma \right)-\xi_{L}^{+}\ln\frac{\xi_{L}^{+}}{\xi^{+}}k_{B}T-\frac{4E^{*}\sqrt{\frac{R_{a}^{2}}{R_{b}}}h_{d}^{3/2}}{3f\left(h_{d}\right)}\right\}+\int_{a}^{\infty }k_{B}T\left(\frac{\partial \xi }{\partial t}\right)\left(1+\ln\frac{\xi }{\xi_{0}}\right)2\pi sds$$
where $\mu =1+\ln\xi /\xi_{0}$ is the local chemical potential of a receptor, $\xi_{L}^{+}\equiv \xi_{L}\left(h_{d}\right)$ and $\xi^{+}\equiv \xi \left(a^{+},t\right)$ denote the receptor–ligand bond density and receptor density, respectively. Noting that $A_{w}=\pi a^{2}=\int_{0}^{h_{d}}f\left(h\right)dh\;$is the contact area, and the function f(h) associated with NP shape can be given as (see Appendix B for details),
(9)$$f\left(h\right)=2\pi \lambda \sqrt{\lambda^{2}{\left(R_{b}-h\right)}^{2}+h\left(2R_{b}-h\right)}\;$$

Receptors on the membrane are driven by the density gradient to the adhesion region, and this driving force is,
(10)$$f_{diffusion}=-\nabla \mu =-\frac{1}{\xi }\frac{\partial \xi }{\partial s}\;$$

Further, the diffusion flux can be given as,
(11)$$j=D\xi f_{diffusion}=-D\frac{\partial \xi }{\partial s}\;$$
with the diffusion coefficient *D*.

Finally, owing to membrane fluidity, the free receptors outside the contact region are mobile within the membrane, and the receptor density can be governed by the diffusion equation as follows [4,52],
(12)$$\frac{\partial \xi }{\partial t}=-\frac{1}{s}\frac{\partial \left(sj\right)}{\partial s}=D\left\{\frac{1}{s}\frac{\partial \xi }{\partial s}+\frac{\partial^{2}\xi }{\partial s^{2}}\right\},\;a\left(t\right)<s<\infty \;$$
when s<a(t), $\xi \left(s,t\right)=\xi_{L}\left(s\right)$, and j(s,t)=0.

The conservation condition of receptors on the membrane can be given as [4,52],
(13)$$\frac{d}{dt}\left[\int_{0}^{a\left(t\right)}\xi_{L}\left(s\right)ds+\int_{a\left(t\right)}^{\infty }\xi \left(s,t\right)ds\right]=0.$$

Substitution of the continuity equation $\frac{\partial \xi }{\partial t}=-\frac{1}{s}\frac{\partial \left(sj\right)}{\partial s}$ into Equation (13) under the boundary conditions $\xi \left(s,t\right)\to \xi_{0}$, j(s,t)→0 as s→∞ yields [4,52],
(14)$$\left(\xi_{L}^{+}-\xi^{+}\right)\frac{da\left(t\right)}{dt}+j^{+}=0\;\mathrm{on}\;s=a\left(t\right),$$
where $j^{+}\equiv j\left(a^{+},t\right)$ denotes the flux in front of the contact edge.

By incorporating the conservation relation and integrating by parts, the integral term in Equation (8) takes the form [4,52],
(15)$$\int_{a}^{\infty }k_{B}T\left(\frac{\partial \xi }{\partial t}\right)\left(1+\ln\frac{\xi }{\xi_{0}}\right)2\pi sds=-2\pi a\frac{da}{dt}k_{B}T\left(\xi_{L}^{+}-\xi^{+}\right)\left(1+\ln\frac{\xi^{+}}{\xi_{0}}\right)-2\pi sk_{B}T\int_{a}^{\infty }\left(-j\right)\frac{\partial \mu }{\partial s}ds\;$$

The first term on the right side represents the energy transport across the front at s=a(t) due to receptor diffusion. The second term on the right in Equation (15) is precisely the rate of energy dissipation as a result of receptor diffusion along the cell membrane.

Similar to the Griffith crack growth criterion, the decreasing rate of free energy should equal to the energy dissipated during receptor diffusion for a power-balanced process [52]. Hence, the energy balance equation at the contact edge features the form,
(16)$$-e_{RL}\xi_{L}{}^{+}+2\kappa H^{2}\left(h_{d}\right)+\gamma +\frac{4E^{*}\sqrt{\frac{R_{a}^{2}}{R_{b}}}h_{d}^{3/2}}{3f\left(h_{d}\right)}-k_{B}T\xi_{L}{}^{+}\ln\frac{\xi^{+}}{\xi_{L}{}^{+}}-k_{B}T\xi_{L}{}^{+}+k_{B}T\xi^{+}=0\;$$

Given the ligand distribution on the NP’s surface, the diffusion equation (12) can be numerically calculated by adopting the finite difference method subjected to the boundary conditions $\xi \left(\infty ,t\right)\to \xi_{0}$ and $\xi^{+}\equiv \xi \left(a^{+},t\right)$. Under the contact edge condition, wherein the contact area reaches the total ellipsoid surface area as $A_{t}=\pi a^{2}=4\pi \left(R_{a}^{2}+2R_{a}R_{b}\right)/3$, the NP is totally internalized, and the wrapping time is determined.

## 3. Results

To simulate the complex situation of non-uniform ligand distribution, we denote a distribution pattern of the ligand density with an approximate harmonic distribution,
(17)$$\xi_{L}=\xi_{L0}\left(1+c\frac{R_{a}}{f\left(h\right)}\cos\frac{\pi h}{R_{b}}\right)\;$$
where the dimensionless constant *c* represents the degree of non-uniformity of ligand distribution. For example, the case of c=0 means that the ligand is uniformly distributed with a density of $\xi_{L0}$. Under such a situation, the total number of ligands is independent of the constant *c* and is denoted by $\xi_{L0}A_{t}$. In Figure 2, we illustrate how such a degree of non-uniformity of the ligand distribution *c*, determines the ligand distribution on an NP with different shapes.

Then, we consider the typical values of the binding energy of a single bond $e_{RL}=15\;k_{B}T$ [55], bending modulus of the cell membrane $\kappa =20\;k_{B}T$ [56], surface tension of the cell membrane $\gamma =0.005\;k_{B}T/{\mathrm{nm}}^{2}$ [22], diffusion coefficient of the receptors on the membrane $D=10^{4}\;{\mathrm{nm}}^{2}/s$ [4,55], and $\xi_{L0}=5000/{\mu m}^{2}$ [4].

### 3.1. Influence of Ligand Distribution on the Cellular Uptake of NPs with Different Shapes

We exclude the effects of the cytoskeletal deformation and set the initial receptor density as $0.01\xi_{L0}$ to systematically investigate the effect of NP shape on uptake. Figure 3 shows the wrapping time as a function of the degree of non-uniformity of ligand distribution, c, for three NPs with different shapes. The wrapping time initially decreases and then increases as the constant c increases; this trend indicates the existence of an optimal ligand distribution corresponding to the minimum wrapping time. This optimal distribution is influenced by the shape of the NP. For spherical NPs, the optimal ligand distribution approaches a uniform distribution, which is consistent with previous predictions by Li et al. [35] and Schubertová et al. [33]. However, when the NP shape changes from spherical to oblate, the optimal ligand distribution transitions from uniform to polarized at the edge of the oblate NP (that is, $h=R_{b}$). For the cellular uptake of elongated NPs, the optimal ligand distribution becomes densely distributed at both ends of the elongated NP. For each case of the oblate and elongated NPs, a large amount of binding energy from the receptor–ligand binding events is required to consume the high bending energy of the local cell membrane with large deformation. Figure 3 also demonstrates that wrapping times are only slightly different for a large range of the constant *c*. When the degree of non-uniformity of ligand distribution, c, is higher or lower than a threshold value, the wrapping process becomes difficult to complete.

### 3.2. Effect of Ligand Distribution on the Cytoskeleton-Associated Endocytosis

We consider the internalization of spherical NPs under the influence of cytoskeleton deformation and determine how the cytoskeleton stiffness may affect the optimal ligand distribution associated with the smallest wrapping time.

Figure 4 displays the relation between the wrapping time and the degree of non-uniformity of ligand distribution, c*,* for spherical NPs with a radius of 50 nm for entry into cells. It can be seen that the cytoskeleton deformation can significantly affect the uptake. In the membrane-mediated endocytosis, the optimal distribution of the ligands coated on the spherical NP surface tends to be uniform. With an increase of Young’s modulus of the cytoskeleton, the optimal ligand distribution becomes dense at the NP ends with a large curvature, in which the large deformation energy of the cytoskeleton plays a key role in resistance to the cellular uptake. When NPs are uptaken by stiff cells, since a large amount of energy for elastic deformation needs to be overcome, long wrapping times will be needed. Similar results for the oblate ellipsoid and elongated NPs can be seen in Figure 5. Regardless, both Figure 4 and Figure 5 show that the range of the ligand distribution constant, c, for effective NP uptake decreases as the Young modulus of the cytoskeleton increases. These findings imply that the internalization of spherical NPs into a soft cytoskeleton can not only shorten the wrapping time but also broaden the range of ligand distribution for effective NP uptake.

### 3.3. Effect of Ligand Distribution on the Cellular Uptake of NPs under Different Initial Receptor Densities

The relation between the wrapping time of NPs and the degree of non-uniformity of ligand distribution, c, for different initial receptor densities is displayed in Figure 6, which further identifies how the ligand distribution may influence the wrapping time under the different initial receptor densities. The semi-axis and engulfing pattern are set as $R_{a}=50\;\mathrm{nm}$ and membrane-mediated endocytosis, respectively. It can be seen that the wrapping time decreases as the initial receptor density increases. Therefore, the role of receptor diffusion becomes negligible for large initial receptor densities. Additionally, there exists a broad range of ligand distribution patterns determined by the coefficient *c*, corresponding to an almost identical wrapping time. The ranges of coefficient *c* are almost independent of the initial density of the receptors. However, the corresponding wrapping time decreases with the increase of the initial density of the receptors.

### 3.4. Effect of Ligand Distribution on the Cellular Uptake of NP with Different Sizes

Size-dependent NP endocytosis has been demonstrated in previous studies [4,5,6]. However, how the ligand distribution and NP size may couple together to influence the cellular uptake of NPs remains unclear. We consider membrane-mediated endocytosis to address this issue. Figure 7 and Figure 8 plot the wrapping time of NPs with different aspect ratios, such as λ=1, λ=5/3, and λ=5/9, as a function of the degree of non-uniformity of ligand distribution c, for NPs with different sizes. The optimal ligand distributions for different NP shapes, such as c=0 (uniform) for a spherical NP, c<0 for an oblate NP, and c>0 for an elongated NP, are determined. The results also show that the shortest wrapping time associated with the optimal ligand distribution increases with an increase in the NP’s size. For large NPs, a long wrapping time is required to recruit numerous mobile receptors to bind with the ligands on the NP’s surface.

For the wrapping of large NPs, there exists a very large range of *c* corresponding to a wrapping time almost identical to the optimal one. Under this situation, resistance to the cellular uptake from membrane deformation becomes negligible when compared to the energy associated with the receptor distribution, and a long wrapping time mainly results from the large area of NPs that should be wrapped.

## 4. Discussion

It has been revealed that the optimal ligand distributions of NPs depend on the NP’s shape and cytoskeleton stiffness in cytoskeleton-associated endocytosis. Such optimal ligand distributions become non-uniform when cytoskeleton deformation is taken into account, which is different from that of membrane-mediated endocytosis on the spherical or cylindrical NPs [35]. The non-uniform optimal ligand distribution results from the competition between the thermodynamic driving force and the kinetics of receptor diffusion. For ligand densities lower than their optimal values at local wrapping edges, insufficient binding energy prolongs the wrapping time. For ligand densities higher than their optimal values at the local wrapping edges, receptor recruitment can also increase the wrapping time, as the high energy dissipation is due to the large change in the configuration entropy of the receptors.

In the cellular uptake of spherical or cylindrical NPs via membrane-mediated endocytosis, the uniform distribution of ligands becomes the optimal one corresponding to the high uptake efficiency. By contrast, during the cellular uptake of ellipsoidal NPs via cytoskeleton-associated endocytosis, the mean curvature of the membrane at each wrapping edge is no longer a constant, but is determined by NP shape. In the power balance in Equation (18), the fourth term at the left hand is the energy contribution of cytoskeleton deformation, which is dependent on the engulfment depth. Therefore, the optimal ligand distribution turns to non-uniform and becomes strongly dependent on NP shape and cytoskeleton stiffness.

Bio-inspired methods from viruses are suitable for designing drug delivery systems. Thus, a biophysical understanding of NP–cell interactions is urgently needed. For spherical NPs, fast wrapping occurs in a large range of different ligand distribution patterns around a uniform distribution; this finding provides a physical insight into robust virus infection rather than the point of view of gene expression [57,58]. The optimal size (tens of nanometers) [4,6] and shape (sphere) [8] have been revealed from a physical optimization standpoint. In this study, we confirm that ligand distribution is another significant factor in determining the receptor-diffusion-mediated NP uptake into cells. The almost uniform ligand distribution of spherical viruses is possibly controlled by physical evolution and guarantees viral infectivity via receptor-mediated endocytosis.

We also examined the critical state as follows:

By solving Equation (16),
(18)$$\xi^{+}=\xi_{L}^{+}\mathrm{ProductLog}\left(\exp\left[1-\frac{\xi_{L}^{+}e_{RL}-2\kappa H^{2}\left(h_{d}\right)-\gamma -4E^{*}\sqrt{R_{a}^{2}/R_{b}}h_{d}^{3/2}/\left(3f\left(h_{d}\right)\right)}{\xi_{L}^{+}k_{B}T}\right]\right).$$
where ProductLog(.) is the Lambert-W function. In the gradient-driven diffusion process of mobile receptors along the cell membrane, effective wrapping requires $\xi^{+}<\xi_{0}$,
(19)$$\xi_{0}>\xi_{L}^{+}\mathrm{ProductLog}\left(\exp\left[1-\frac{\xi_{L}^{+}e_{RL}-2\kappa H^{2}\left(h_{d}\right)-\gamma -4E^{*}\sqrt{R_{a}^{2}/R_{b}}h_{d}^{3/2}/\left(3f\left(h_{d}\right)\right)}{\xi_{L}^{+}k_{B}T}\right]\right).$$

When the receptor density in front of the contact edge $\xi^{+}$ is equal to the initial receptor density $\xi_{0}$, the wrapping process begins to be terminated. Accordingly, the solution to Equation (18) in such a critical case, $\xi^{+}=\xi_{0}$, yields the critical ligand density in front of the contact edge,
(20)$$\xi_{L}^{+}{|}_{max}=\frac{k_{B}T\xi_{0}-[2\kappa H^{2}\left(h_{d}\right)+\gamma +4E^{*}\sqrt{R_{a}^{2}/R_{b}}h_{d}^{3/2}/\left(3f\left(h_{d}\right)\right)]}{k_{B}T\mathrm{ProductLog}\{\frac{e^{1-\frac{e_{RL}}{k_{B}T}}}{k_{B}T\xi_{0}}[k_{B}T\xi_{0}-[2\kappa H^{2}\left(h_{d}\right)+\gamma +4E^{*}\sqrt{R_{a}^{2}/R_{b}}h_{d}^{3/2}/\left(3f\left(h_{d}\right)\right)]]\}}.$$

If the ligand density at the wrapping edge is larger than the critical value, then the wrapping process cannot occur due to the large unfavorable entropic contribution of the energy from the receptor configuration change.

Our study presents certain limitations. Although we provide a detailed description on the dependence of the optimal ligand distribution on the NP shape and cytoskeleton stiffness, we have treated the NPs as a rigid body. We did not consider in detail whether NP deformation significantly influences the optimal ligand distribution. We also disregard membrane tension [18], the kinetic reaction between receptor and ligand molecules [32,59,60,61], as well as the NP concentration [62]. We apply these assumptions in this study so that we can easily focus on the effect of ligand distribution on the cellular uptake of ellipsoidal NPs via cytoskeleton-associated endocytosis.

## 5. Conclusions

We show the systematically distinct effects of ligand distribution on the dynamics of cytoskeleton-associated endocytosis of ellipsoidal NPs with different sizes under different initial receptor densities. NPs with the same number of ligands but different distributions can have a wide range of different dynamic processes of cellular uptake. We find that there exist optimal ligand distributions corresponding to the minimum wrapping times for NPs with different shapes. Unlike the findings in previous studies on the cell entry of spherical and cylindrical NPs with different ligand distributions via membrane-mediated endocytosis, such optimal ligand distributions become non-uniform and dependent on the NP shape and cytoskeleton stiffness. For example, the optimal distribution favors that the ligands are densely distributed in the region with large local curvature or large cytoskeleton deformation. These findings supply an insight into the physical understanding of NP–cell interactions and may provide guidelines for targeted drug delivery.

## Figures and Tables

**Figure 1 membranes-11-00993-f001:**
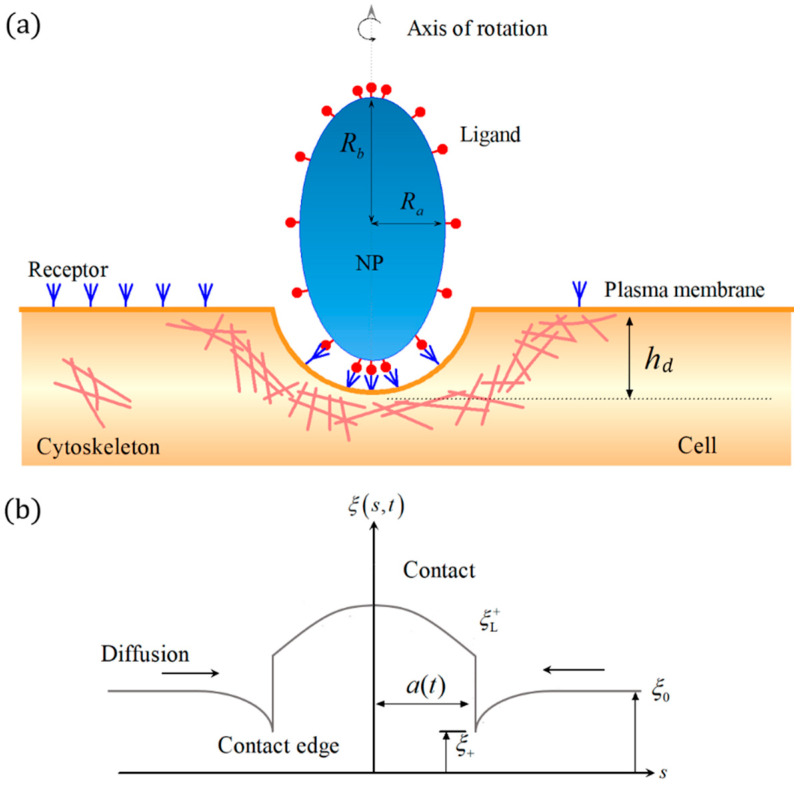
Schematic of the interaction between ellipsoidal NPs with different ligand distributions and cells via cytoskeleton-associated endocytosis. (**a**) Remote mobile receptors diffusing to binding sites to drive cellular uptake. (**b**) Receptor density distribution along the cell membrane.

**Figure 2 membranes-11-00993-f002:**
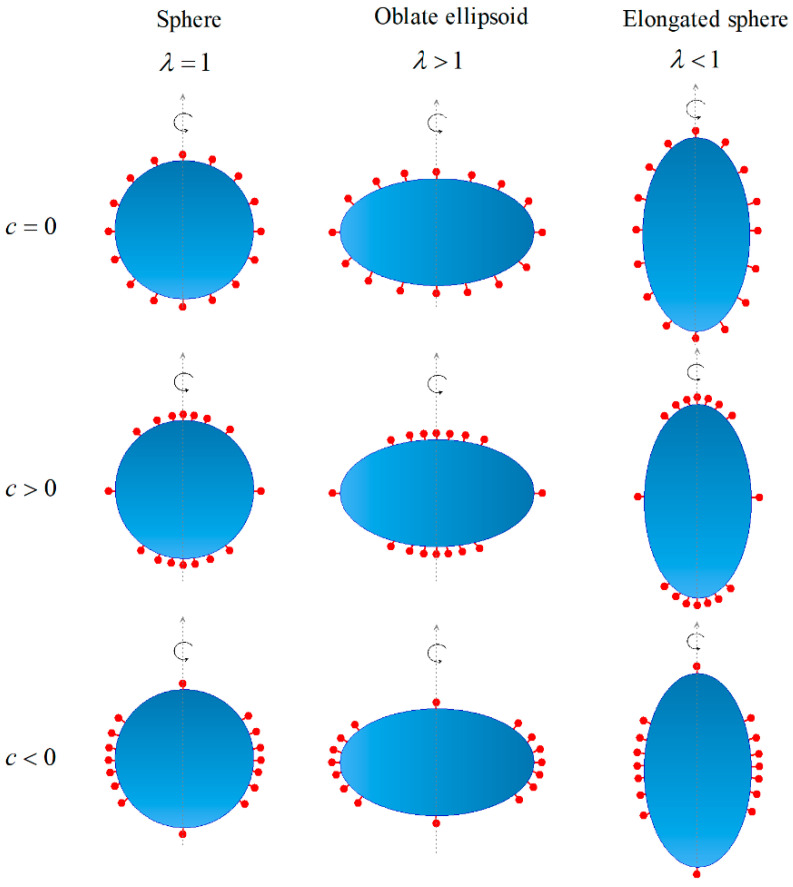
Schematic on how the dimensionless constant *c* determines the ligand pattern on the surfaces of NPs with different shapes.

**Figure 3 membranes-11-00993-f003:**
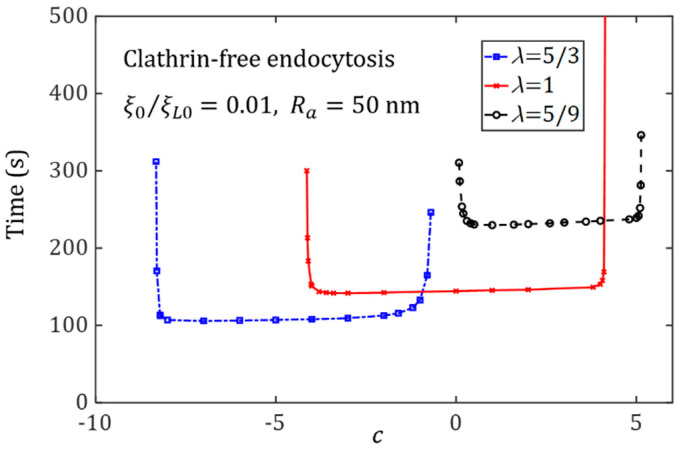
In the cellular uptake of NPs with $R_{a}=50\;\mathrm{nm}$ via membrane-mediated endocytosis, wrapping time as a function of the degree of non-uniformity of ligand distribution, *c*, for$\;\xi_{0}=0.01\xi_{L0}$ and different ratios λ=5/3, 1 and 5/9.

**Figure 4 membranes-11-00993-f004:**
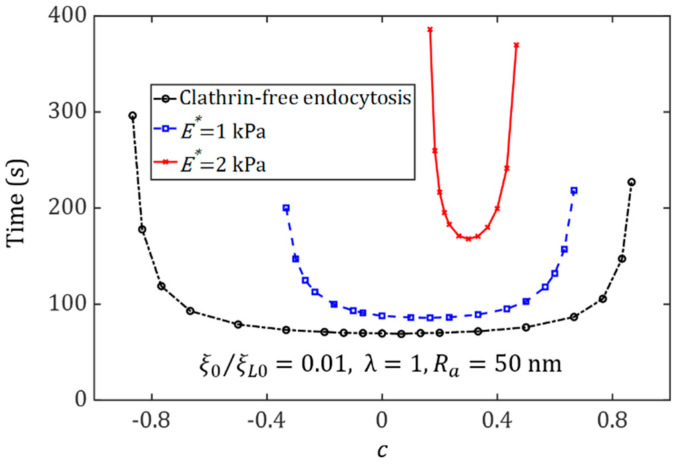
Wrapping time as a function of the degree of non-uniformity of ligand distribution, *c*, for membrane-mediated endocytosis and cytoskeleton-associated endocytosis pathways with $E^{*}=1\;\mathrm{kPa}$, $E^{*}=2\;\mathrm{kPa}$, during the cellular uptake of spherical NPs ($R_{a}=50\;\mathrm{nm}$, λ=1) with $\xi_{0}=0.01\xi_{L0}$.

**Figure 5 membranes-11-00993-f005:**
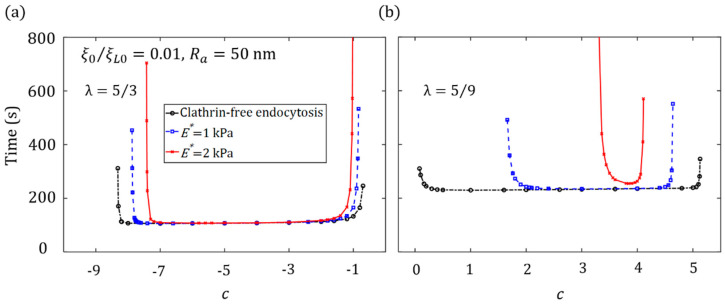
Wrapping time as a function of the degree of non-uniformity of ligand distribution, *c*, for membrane-mediated endocytosis and cytoskeleton-associated endocytosis pathways with $E^{*}=1\;\mathrm{kPa}$, $E^{*}=2\;\mathrm{kPa}$, during the cellular uptake of (**a**) oblate ellipsoid (λ=5/3) and (**b**) elongated NPs (λ=5/9) with $\xi_{0}=0.01\xi_{L0}$.

**Figure 6 membranes-11-00993-f006:**
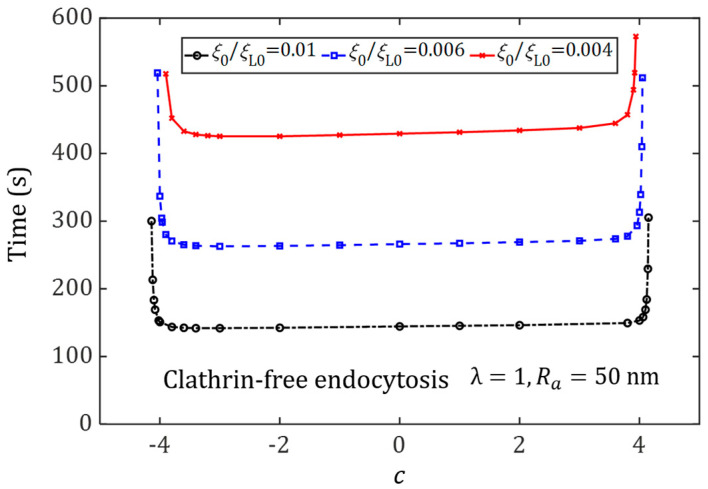
Wrapping time as a function of the degree of non-uniformity of ligand distribution, *c*, at different initial receptor density situations $\xi_{0}=0.004\xi_{L0},\;\xi_{0}=0.006\xi_{L0}$ and$\;\xi_{0}=0.01\xi_{L0}$, during the cellular uptake of spherical NPs ($R_{a}=50\;\mathrm{nm}$, λ=1) for the membrane-mediated endocytosis pathway.

**Figure 7 membranes-11-00993-f007:**
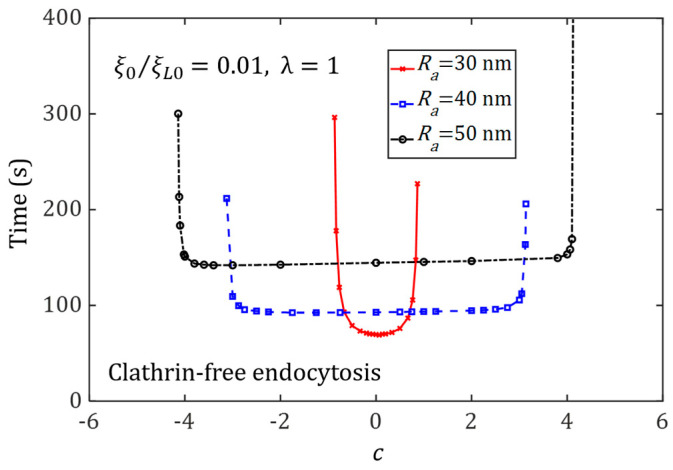
Wrapping time as a function of the degree of non-uniformity of ligand distribution, *c*, for different NP sizes, during the cellular uptake of spherical NPs (λ=1) via membrane-mediated endocytosis, with $\xi_{0}=0.01\xi_{L0}$.

**Figure 8 membranes-11-00993-f008:**
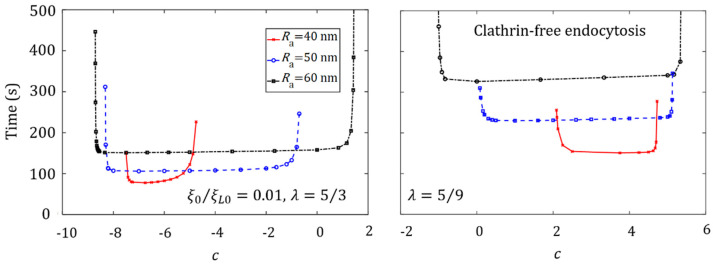
Wrapping time as a function of the degree of non-uniformity of ligand distribution, *c*, for different NP sizes, during the cellular uptake of oblate ellipsoid (λ=5/3) and elongated NPs (λ=5/9) via membrane-mediated endocytosis, with$\;\xi_{0}=0.01\xi_{L0}$.

## Data Availability

Not applicable.

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
