# Peer review of "Role of Ligand Distribution in the Cytoskeleton-Associated Endocytosis of Ellipsoidal Nanoparticles"

_membranes, 2021, doi:10.3390/membranes11120993_

Round 1

Reviewer 1 Report

The comments are in the attached file

Author Response

We thank the referee for his/her constructive comments. The paper has been revised accordingly. Enclosed please find the itemized response to each specific comment.

Reviewer 2 Report

In this work by Zhang et al., the authors have studied the role of ligand distribution in the cytoskeleton-associated endocytosis of ellipsoidal NPs. Using the elasticity–diffusion model the authors found that an optimal ligands distribution on nanoparticles surface is required for an efficient cellular uptake process. The authors present their data with well-organized figures which, for the most part, are both visually appealing and clear.

However, as the authors have published very similar works recently (such as PMID: 28573012), I believe that the present work does not provide any potential improvements to the field. While I am not intimately familiar with the field, I wonder if those who are would find this work interesting.

Author Response

(The authors gave the same response as above.)

Reviewer 3 Report

Reviewing of the paper “Role of ligand distribution in the cytoskeleton-associated endocytosis of ellipsoidal nanoparticles”

This manuscript describes the development of a theoretical dynamic model to investigate the influence of ligand distribution on the cellular uptake of ellipsoidal NPs, driven by the ligand-receptor interaction.

The same research team has already published their results on the effects of ligand distribution on receptor-mediated cellular uptake of cylindrical nanoparticles, using a similar approach.

Results showed distinct effects of ligand distribution on the dynamics of endocytosis of NPs with different sizes on respect to different initial receptor densities.

The originality of the work and the scientific relevance can be considered of a good level, considering that other papers are already present in literature describing more specific cases (for instance Schubertová et al. Influence of ligand distribution on uptake efficiency. Soft Matter 2015, 449 11, 2726-2730) as mentioned by the authors themselves.

The manuscript appears to be well organized, with good acknowledgement of the work of others in the references and perfectly fits the aim of the special issue “Biophysics and Mechanics of Cell Membranes”.

As stated by the authors at the end of the Discussion chapter, this study presents some limitations and approximations due to the complexity of the topic treated.

One point not been considered is the influence of NPs concentration on the NP-cell interaction, an aspect of sure interest but really hard to be integrated in the mathematical traction of this research. I hope the authors will continue on this research integrating also this and the other aspects they excluded from their model, in order to better clarify the interaction between nanomaterials and cells.

Anyway, I think this paper can be Accepted in the Present Form.

Author Response

(The authors gave the same response as above.)

Round 2

Reviewer 2 Report

Thanks for modifying the manuscript.